# Simplified clinical algorithm for identifying patients eligible for immediate initiation of antiretroviral therapy for HIV (SLATE): protocol for a randomised evaluation

Sydney Rosen,[1,2] Matthew P Fox,[1,2,3] Bruce A Larson,[1] Alana T Brennan,[1,2] Mhairi Maskew,[2] Isaac Tsikhutsu,[4] Margaret Bii,[4] Peter D Ehrenkranz,[5] WD Francois Venter[6]

For numbered affiliations see end of article.

**Correspondence to**
Professor Sydney Rosen;
sbrosen@bu.edu

## ABSTRACT

**Introduction** African countries are rapidly adopting guidelines to offer antiretroviral therapy (ART) to all HIV-infected individuals, regardless of CD4 count. For this policy of 'treat all' to succeed, millions of new patients must be initiated on ART as efficiently as possible. Studies have documented high losses of treatment-eligible patients from care before they receive their first dose of antiretrovirals (ARVs), due in part to a cumbersome, resource-intensive process for treatment initiation, requiring multiple clinic visits over a several-week period.

**Methods and analysis** The Simplified Algorithm for Treatment Eligibility (SLATE) study is an individually randomised evaluation of a simplified clinical algorithm for clinicians to reliably determine a patient's eligibility for immediate ART initiation without waiting for laboratory results or additional clinic visits. SLATE will enrol and randomise (1:1) 960 adult, HIV-positive patients who present for HIV testing or care and are not yet on ART in South Africa and Kenya. Patients randomised to the standard arm will receive routine, standard of care ART initiation from clinic staff. Patients randomised to the intervention arm will be administered a symptom report, medical history, brief physical exam and readiness assessment. Patients who have positive (satisfactory) results for all four components of SLATE will be dispensed ARVs immediately, at the same clinic visit. Patients who have any negative results will be referred for further clinical investigation, counselling, tests or other services prior to being dispensed ARVs. After the initial visit, follow-up will be by passive medical record review. The primary outcomes will be ART initiation ≤28 days and retention in care 8 months after study enrolment.

**Ethics and dissemination** Ethics approval has been provided by the Boston University Institutional Review Board, the University of the Witwatersrand Human Research Ethics Committee (Medical) and the KEMRI Scientific and Ethics Review Unit. Results will be published in peer-reviewed journals and made widely available through presentations and briefing documents.

**Trial registration** NCT02891135

## Strengths and limitations of this study

► Evaluates a practical, simple algorithm for accelerating initiation of HIV treatment, a point at which many patients are currently lost from care.
► Will contribute to the evidence base for achieving global targets.
► Individually randomised trial conducted in six study sites in two African countries, making the results generalisable to the region.
► Comparison arm is standard care, which could change over the course of the study.

## INTRODUCTION

In its 2015 revision of the global guidelines for HIV care and treatment, the WHO called for initiating lifelong antiretroviral therapy (ART) for all patients testing positive for HIV, rather than waiting for a patient's CD4 count to fall below 500 cells/mm$^3$, as in the previous guidelines.[1] Many African countries have adopted this recommendation and are rapidly developing new instructions and procedures for its implementation.

In its recommendation, which is referred to as 'treat all' or 'test and treat', the WHO cited 'increases in ART uptake and linkage to care, reduction in the time between HIV diagnosis and ART initiation regardless of baseline CD4 cell count and an increase in the median CD4 value at ART initiation' as some of the main benefits to be gained from the guideline change, a conclusion recently confirmed in a modelling analysis.[2] For countries to benefit from treat all, therefore, millions of new patients must be initiated on ART as efficiently as possible, while ensuring that patient autonomy, welfare and commitment to lifelong ART are not jeopardised by

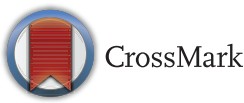

the initiation process. 'Treat all', by removing the requirement of a preinitiation CD4 count, promises to simplify and accelerate treatment initiation, if efficient procedures can be developed and implemented. Studies from throughout sub-Saharan Africa have documented high losses of treatment-eligible patients from care after they test positive for HIV but before they receive their first dose of antiretrovirals (ARVs) due to a wide range of facility-level and patient-level barriers to initiation.[3-4] Multiple preparatory visits, long waiting times, stock outs of supplies, staff absences and poor communication between staff and patients all deter treatment initiation.[5–8]

We previously proposed a simplified clinical algorithm for allowing nurses and other clinicians to reliably determine a patient's eligibility for immediate ART initiation without waiting for laboratory results or additional clinic visits.[9] The Simplified Algorithm for Treatment Eligibility (SLATE) includes a symptom report, medical history, brief physical exam and readiness assessment; patients who 'screen in' on all four screens can be dispensed ARVs on the spot, without any further steps, while those who do not are referred for further assessment, care or support. An international technical consultation in 2015 regarded evaluating this algorithm as the highest priority question on the operational research agenda it developed.[9]

In this paper, we describe the protocol for a pragmatic, individually randomised evaluation of the effectiveness of the new algorithm in increasing ART uptake and retention in care in South Africa and Kenya. The SLATE study will evaluate whether patients who are administered the SLATE screens and either initiated on ART immediately or referred for further care have higher uptake of ART at 28 days and better retention on ART at 6 months than do patients who receive standard ART initiation.

The purpose of the SLATE algorithm is to allow clinics to initiate ART in a way that minimises the time required for both patients and staff and reduces loss to follow-up prior to treatment initiation for the majority of patients who require no additional clinical care or psychosocial support before they can start. For those who would benefit from additional care or support—whether an additional counselling session, a tuberculosis (TB) test or a more comprehensive physical examination—the algorithm is intended to identify and refer such patients for the additional care required, without slowing down the process for those who do not.

## METHODS AND ANALYSIS

SLATE will be an individually randomised, non-blinded, pragmatic evaluation of the effect of the SLATE algorithm on ART initiation and retention in care, conducted at three public sector clinics each in Kenya and South Africa.

### Intervention: the SLATE algorithm and immediate ART initiation

The intervention to be evaluated in this study is a clinical algorithm that streamlines and structures the information required from patients before ARVs are dispensed

for the first time. The algorithm consists of four 'screens': symptom report, medical history, brief physical examination and readiness assessment. Patients who 'screen in'—have satisfactory responses to all four screens—can then be dispensed ARV medications on the spot, without any further steps or delays. Initiation of ART immediately after completing the four SLATE screens, without any further services required, is labelled 'immediate' initiation in this study. Patients who 'screen out'—have at least one unsatisfactory response on a screen—are referred for further services, such as a laboratory test, more intensive physical examination or counselling, before ARVs are dispensed.

Table 1 and figure 1 describe the SLATE screens in detail, and the questions from the SLATE screening instruments to be used in the study are available in supplementary file 1. Most of the information collected by the screens is consistent with current practice in study countries. The fourth screen, which aims to assess patient 'readiness' to start ART, addresses an issue for which there is little standardisation among countries or clinics. While emotional and psychological readiness to start ART is often regarded as important to achieving good medication adherence after initiation, a published review of readiness instruments for HIV initiation found that none was notably successful in predicting patient readiness for ART, as indicated by adherence once on treatment.[10] Primarily to ensure that patients have an opportunity to express any reservations they may have about starting ART and ask questions, we developed a three-question readiness assessment instrument.

### Outcomes, randomisation and sample size

The study has two primary outcomes, both based on the outcomes proposed for operational research on ART initiation in a previous publication.[9] For primary outcome 1, we will estimate the proportion of patients in each arm (standard care and intervention) who initiate ART within 28 days of study enrolment. Previous studies have found that 28 days is a sufficient time interval for a majority of patients found eligible for ART to complete the steps required to start treatment under routine care.[11 12] A patient who has not initiated within 28 days will be regarded as failing to achieve this primary outcome. We note that 28 days is a relatively generous interval to allow for achievement of this outcome, since the intervention is designed to allow ART initiation within 1 day (ie, on the same day). From the perspective of a patient who is asymptomatic at first clinic visit and thus eligible for the SLATE algorithm , however, there appear to be few clinical benefits to starting within 1 day compared with 28 days. The real benefit of the SLATE approach is to eliminate the possibility of loss to follow-up between the first visit and treatment initiation. By providing a relatively long interval for standard arm patients to start ART, we can be assured that any observed differences between the arms reflect true benefits to patients' health. Secondary outcomes, as described below, will also allow proportions

**Table 1** SLATE algorithm screens

| Screen | Overall purpose of screen | Reasons for screening out | Justification | If screen out, anticipated next step |
|---|---|---|---|---|
| Symptom report | Identify self-reported conditions that require additional investigation | Current cough, fever, night sweats or recent weight loss | These symptoms comprise the WHO-recommended TB symptom screen[16] | Referral for TB test |
| | | Persistent headache for >2 days | Symptom of cryptococcal meningitis[17 18] | Referral for CrAg screening |
| | | Other self-reported symptoms | Other symptoms could indicate the need for further clinical investigation | Referral for additional clinical consultation |
| Medical history | Through self-report identify individuals on concurrent medications or who may struggle with adherence | On ART previously | Patients who have been on ART in the past may require additional adherence counselling | Referral for additional counselling session |
| | | Started TB treatment within the past 2 weeks | Guidelines recommend a 2-week delay in ART initiation for patients starting TB treatment | Appointment for ART initiation immediately after the 2-week window has passed |
| | | Concurrent medications for epilepsy or current warfarin | These medications can interact with ARVs | Referral for additional clinical or pharmacy consultation |
| | | Current substance abuse | Use of recreational drugs or overuse of alcohol can create challenges for chronic medication adherence | Referral for additional counselling session |
| Physical examination | Record weight, height, temperature and blood pressure and identify any observable conditions that require additional investigation | Any conditions that call for further investigation prior to ART initiation | Patient may identify previously unreported symptoms or clinician may observe conditions that indicate a need for further clinical investigation before starting ART | Referral for additional clinical consultation |
| Readiness assessment | Confirm that the patient feels ready to start ART today | Responses that indicate reluctance, hesitation or concerns in starting and adhering to treatment | Creates a structured opportunity for clinician and patient to discuss any concerns that the patient has not yet raised | Referral for additional counselling and follow-up support as indicated |

ART, antiretroviral therapy; CrAg, cryptococcal antigen; SLATE, Simplified Algorithm for Treatment Eligibility; TB, tuberculosis.

of patients initiating ART at any interval after study enrolment to be assessed.

For primary outcome 2, we will estimate the proportion of patients in each arm who initiate ART within 28 days (ie, achieve primary outcome 1) and are alive, in care and retained on ART 8 months after study enrolment. Eight months was selected to allow up to 1 month (28 days) to initiate ART, 6 months of follow-up after treatment initiation and up to 1 month to return for the 6-month routine clinic visit. Although viral suppression is preferable to retention in care as a measure of ART success, we are not confident that all the study sites will consistently perform routine viral load tests at 6 months, and viral suppression will be considered as a secondary outcome. To allow for the irregularity of clinic visits, we will allow any clinic visit between 5 and 7 months after treatment initiation (or between 6 and 8 months after study enrolment, taking into account the month allowed for treatment initiation) to represent the 6-month visit. To help explain our results, we will also describe reasons for not achieving this primary

outcome to the extent that this information is available in routine clinic records and will note any adverse events encountered during the 8-month follow-up period for this outcome.

Secondary outcomes are described in table 2.

Study patients will be randomised 1:1, using block randomisation in blocks of six, to the intervention arm or standard care arm. Randomisation will be done at the country level, with an expected sample size of 160 subjects per site but with the possibility of allocating more or fewer slots to individual sites, while maintaining randomisation rules, if enrolment rates differ. Blinding is not possible in a pragmatic evaluation such as SLATE, as each arm will follow very different procedures post-randomisation.

The target sample size for the study, which was powered for the second primary outcome (initiated ART within 28 days and alive, in care and retained on ART 8 months after study enrolment), is 480 patients per country or 960 in total. Using results of the RapIT randomised trial we recently conducted in South Africa,[11] we estimated that

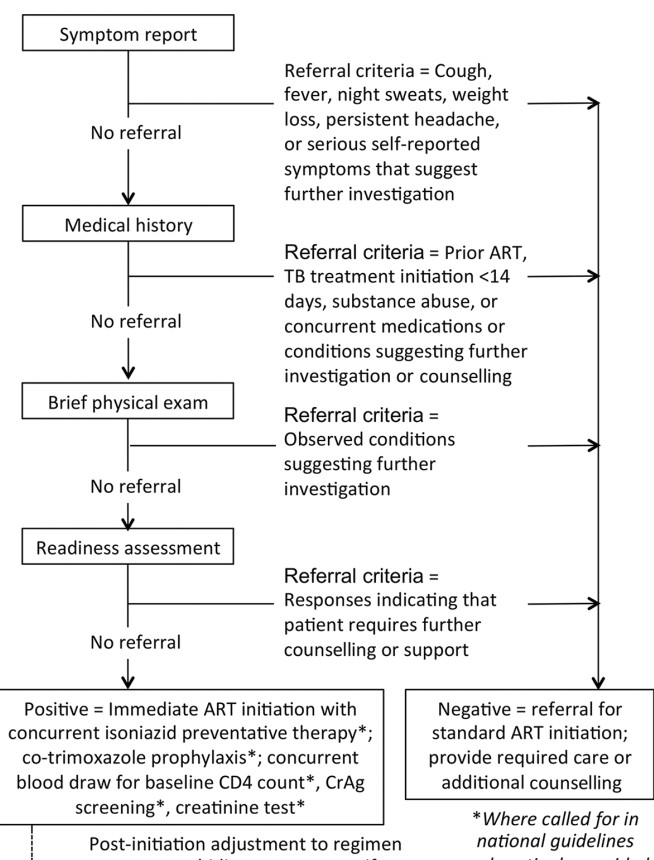

**Figure 1** Adapted from Rosen S, Fox MP, Larson B, Sow PS, Ehrenkranz PD, Venter F, Manabe Y, Kaplan J, for the Models for Accelerating Treatment Initiation Technical Consultation. Accelerating the uptake and timing of antiretroviral therapy initiation in sub-Saharan Africa: an operations research agenda. *PLoS Med* 2016; 13: e1002106. DOI:10.1371/journal.pmed.1002106 (CC BY 4.0). ART, antiretroviral therapy; CrAg, cryptococcal antigen; TB, tuberculosis.

65% of treatment-eligible patients will be initiated on antiretroviral therapy and retained on ART in the standard arm, and we considered an increase to 80% to be programmatically important. Using an α of 0.05, power of 90%, 1:1 randomisation and an uncorrected Fisher's exact test, this requires a minimum sample size of 197 patients per group, which we will increase to a maximum of 240 to ensure sufficient power, with stratification by country, if patients withdraw or are found ineligible after consent.

## Study sites and ethics review

South Africa and Kenya, where the study will be conducted, differ widely in income and HIV prevalence, and thus will allow us to determine the effectiveness of the intervention in varying settings. In each country, patients will be enrolled at three public sector clinics that have infrastructure and staff that are typical of the facilities that treat the vast majority of HIV patients. The only restriction we applied in selecting sites was that they have relatively high

patient volumes, to speed up study enrolment. The sites are described in table 3.

Study implementation in each country will be led by a local research organisation subcontracted to the primary study implementer, Boston University (table 3). At each site, the study will employ a study clinician in the same professional cadre that initiates patients onto ART under standard care. In South Africa, ART is typically initiated by public health nurses, who are the most senior cadre of nurse in that country; in Kenya, ART is initiated by clinical officers. The study clinicians will be trained on study procedures but have no additional clinical training or qualifications, beyond what is typical in routine clinic settings. Each site will also have one or two study assistants, who will be responsible for screening potential subjects, obtaining informed consent, administering the baseline questionnaire and performing other non-clinical tasks. The study will have two dedicated rooms or private spaces at the sites—one for the nurse/clinical officer and one for the study assistant. Where necessary, we will renovate existing space or provide a separate, stand-alone space, such as a trailer, for study activities.

The study protocol, which is available as supplementary file 2, has been approved by the Boston University Institutional Review Board and the University of the Witwatersrand Human Research Ethics Committee (Medical) and is under review by the KEMRI Scientific and Ethics Review Unit. It is registered with Clinicaltrials. gov as NCT02891135.

## Screening and enrolment

At each site, we will recruit 160 adult patients (≥18 years) who have tested positive for HIV, either at the current clinic visit or previously, and have not yet initiated ART. Pregnant women will be excluded, as procedures for initiating and managing pregnant women on ART differ from those for non-pregnant adults. Patients who intend to receive further HIV care at a different clinic, rather than the study site, will also be excluded, as will those who are determined by study staff to be physically, mentally or emotionally unable to participate. We note that these study eligibility criteria will allow enrolment of patients at varying points in the HIV care cascade. Some will be enrolled a few moments after having their first positive HIV test, while others will have been attending pre-ART monitoring visits for several years. The study sample will thus reflect the full range of ART-eligible patients presenting at African clinics as treat-all guidelines are rolled out but will face heterogeneity in patient characteristics within each arm.

Site staff will refer patients with HIV to the study assistant for study screening and consent. Patients will be screened consecutively in the order in which they are referred by the site staff on a first-come, first-served basis, based on the availability of the study assistant(s). While some patients may prefer not to wait until the study assistant is available and opt out of being screened for enrolment, we do not anticipate that the screening

**Table 2** Secondary outcomes

| Secondary outcome | Justification and/or further description | Data analysis |
|---|---|---|
| **ART outcomes** | | |
| ART initiation within 14 days of study enrolment | National guidelines in both study countries[19 20] recommend initiation ≤14 days | Intention-to-treat analysis; comparison of proportions between groups presented as a risk difference and 95% CIs |
| Time to initiation, in days | One goal of SLATE algorithm is to accelerate initiation; time to initiation captures any effect on this. | Intention-to-treat analysis; comparison of time to initiation presented as survival curves with log rank test |
| Viral suppression by 8 months after study enrolment | Allows ≤1 month (28 days) to initiate ART, 6 months of follow-up after treatment initiation and ≤1 month to return for the 6-month routine clinic visit | Intention-to-treat analysis; comparison of proportions between groups presented as a risk difference and 95% CIs. Reasons for not achieving this outcome will also be described to the extent that routinely collected follow-up data allow. |
| Retention in care 14 months after study enrolment | Allows ≤1 month (28 days) to initiate ART, 12 months of follow-up after treatment initiation and ≤1 month to return for the 12-month routine clinic visit; any visit 12–14 months after study enrolment will represent the 12-month visit | Intention-to-treat analysis; comparison of proportions between groups presented as a risk difference and 95% CIs |
| Retention in care at 16 months after study enrolment | Allows ≤1 month (28 days) to initiate ART, 12 months of follow-up after treatment initiation and ≤3 months to return for the 12-month routine clinic visit, to allow comparability with other studies of 12-month retention in care, which often define loss to follow-up as 90 days late for the last scheduled visit. | Intention-to-treat analysis; comparison of proportions between groups presented as a risk difference and 95% CIs |
| **SLATE evaluation** | | |
| Proportions of study patients who screen in and screen out for immediate ART initiation using SLATE algorithm criteria | Will provide guidance on proportions of patients who could be initiated under SLATE if adopted as routine care | Intention-to-treat analysis; comparison of proportions between groups presented as a risk difference and 95% CIs |
| Reasons for ineligibility | Will provide guidance on types of referral services required from clinics | Descriptive analysis of proportions of patients screening out for each possible reason indicated on SLATE screens |
| Patient preferences on the speed and timing of ART initiation | Baseline questionnaire data | Descriptive analysis of medians and IQRs for continuous outcomes and proportions and corresponding 95% CIs for categorical outcomes |
| **Health system outcomes** | | |
| Costs to patients of ART initiation under standard and intervention procedures | SLATE is hypothesised to reduce the number of clinic visits required for ART initiation and thus costs to patients | Sum of clinic visit costs and time spent from enrolment visit to visit at which ARVs are dispensed, calculated from questionnaire responses. |
| Costs to providers of ART initiation under standard and intervention procedures and cost-effectiveness of intervention | SLATE is hypothesised to reduce the number of clinic visits required for ART initiation and thus costs to providers | Estimate of provider costs using previously described[21] bottom-up costing methods, with resource utilisation extracted from medical records and CRFs and unit costs obtained from study sites. The average cost to the provider per patient achieving each primary outcome will be compared between intervention and standard initiation groups to provide an estimate of the cost-effectiveness of the two strategies. Costs will be reported as means (SD) and medians (IQRs) in local currencies and US dollars. |

ART, antiretroviral therapy; CRFs, case report forms; SLATE, Simplified Algorithm for Treatment Eligibility.

**Table 3** Study sites

| Country, implementer and registry | Province/district or county | Site name | Patient population served |
|---|---|---|---|
| South Africa: Health Economics and Epidemiology Research Office (HE²RO), University of the Witwatersrand, Johannesburg. South African National Health Research Database GP_2016RP42_318. | Gauteng Province, Johannesburg Metro | OR Tambo Primary Health Clinic | Urban informal settlement (Diepsloot) |
| | Gauteng Province, Johannesburg Metro. | Alexandra Community Health Centre | Urban informal settlement (Alexandra) |
| | Gauteng Province, Ekhurhuleni Metro | Jabulani Dumani Community Health Centre | Urban informal settlement (Vosloorus) |
| Kenya: Kenya Medical Research Institute/ Walter Reed Projects (KEMRI/WRP), Kericho. Pan African Clinical Trial Registry PACTR201609001783150. | Kericho County | Kericho County Referral Hospital | Kericho town and surrounding rural areas |
| | Nandi County | Kapsabet County Referral Hospital | Kabsabet town and surrounding rural areas |
| | Kisumu County | Kombewa County Hospital | Rural areas northwest of Kisumu |

process will introduce any biases to the sample ultimately enrolled. All patients who are found eligible for the study through the screening process will be asked for written informed consent and enrolled in the study. Following consent, female patients will be asked to complete a pregnancy test, and any who are found to be pregnant will be withdrawn from the study and escorted to the site's antenatal clinic to enrol in antenatal care and prevention of mother-to-child transmission care.

One of the goals of implementation science studies is to generate evidence in a time frame that is programmatically relevant. To accelerate reaching the country enrolment targets, unequal enrolment by site will be allowed, to take advantage of sites that have higher patient volumes. Enrolment in SLATE is expected to be completed within a 6-month to 8-month period in 2017. The first primary outcome can then be estimated just 28 days after the last patient has been enrolled, while the second primary outcome will require 8 months of follow-up. We thus expect most study results to be available by the end of 2018.

## Procedures

Study procedures are illustrated in figure 2. Prior to randomisation, patients who provide written informed consent to study participation will be administered a baseline questionnaire by the study assistant, with sections on demographic and socioeconomic status, costs incurred per clinic visit and preferences for the timing and speed of ART initiation. Randomisation assignments, stored in sealed envelopes, will be opened and conveyed to participants by the study assistant, after completing the questionnaire. After randomisation, patients assigned to the standard arm will have no further study activities. They will receive a payment equivalent in value to US$5–15 to thank them for participating and compensate them for their time. Payment will be in the form of a shopping voucher that can be used at nearby grocery/ general goods stores in South Africa and cash in Kenya. The study assistant will then escort standard arm patients

back to their original place in the patient queue and complete a standard clinic visit, as they would have done in the absence of the study, and will follow the clinic's regular schedule of procedures and visits for ART initiation. They will have no further interaction with study staff after this point.

Patients assigned to the intervention arm will be introduced to the study nurse/clinical officer, who will administer the four SLATE screens. For the study, all four screens will be completed for all patients assigned to the intervention arm, so that we can obtain a complete data set for all patients. If SLATE were used in routine care, we expect that the screens would be administered sequentially, so that a negative response to the first screen (eg, report of a symptom of TB) would allow a decision to refer the patient for additional care to be made without having to complete the second screen, and so on. Patients would progress to the next screen only after being found positive on the previous one, where 'positive' refers to satisfactory responses to all items on the screen, while 'negative' indicates a response that requires referral. We expect that for the vast majority of patients, each screen will take less than 5 min to administer. We would anticipate that in routine practice the algorithm will take 15–20 min for most patients.

Following administration of the four screens, intervention arm patients will have a blood draw to allow the study to collect baseline CD4 counts and for any preinitiation laboratory tests called for under standard care. Blood samples will be sent for processing at the same laboratories used by the study sites. Although baseline CD4 counts are no longer used in either study country to establish ART eligibility, they continue to be performed for all patients initiating ART in both countries. Baseline CD4 count remains a strong indicator of early outcomes on ART and is therefore an important variable for the study analysis.

The results of the four screens will indicate whether a patient has either 'screened in' or 'screened out' of the

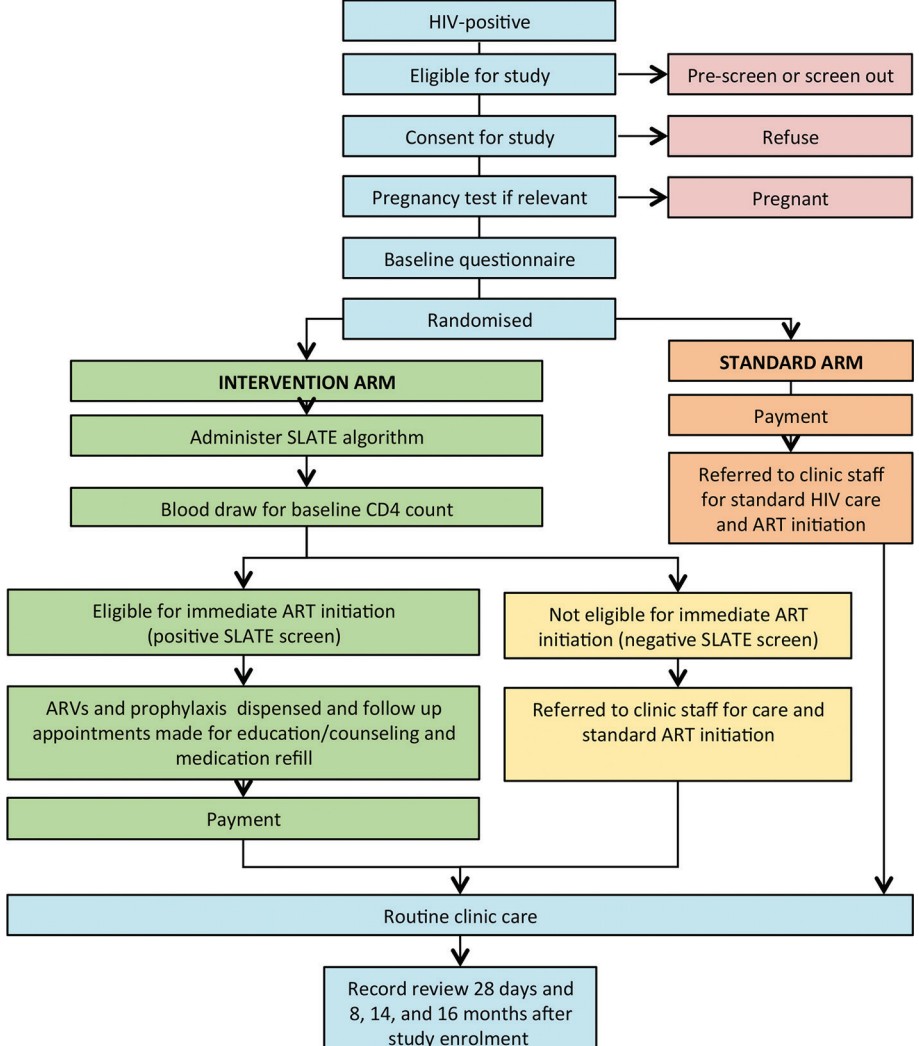

**Figure 2** Study flow diagram. ART, antiretroviral therapy; ARV, antiretroviral; SLATE, Simplified Algorithm for Treatment Eligibility.

SLATE algorithm. Findings in any of the four screens that suggest that further clinical investigation, counselling, laboratory tests or other services are advisable prior to dispensing ARVs will cause a patient to screen out. Patients who screen out will be referred for the suggested follow-up services and escorted to the appropriate clinic location for immediate follow-up at the same visit where possible. If the recommended follow-up service is not available immediately, a follow-up appointment will be made for the patient.

It is important to note that screening out of the SLATE algorithm will not necessarily preclude same-day ART initiation, either in the study or if the algorithm were used in routine care. Many patients who screen out will receive the additional service at the same clinic visit (eg, an additional meeting with a counsellor) and can still be prescribed ARVs before completing the visit. In fact, we expect that many patients who screen out of SLATE could indeed start ART on the same day, if the clinic has the capacity to provide the follow-up service that is needed and there is enough time left in the day for the patient to be served. SLATE is not

intended to delay ART initiation for those who screen out, but rather to accelerate the process for those who screen in.

For patients who 'screen in'—have satisfactory responses to all four SLATE screens, do not require any additional services and are eligible for immediate dispensing of ARVs—the study clinician will then have a brief conversation with the patient to confirm that the patient remains ready to start ART, understands what happens next in the study and has no further questions or concerns. The clinician will then write a prescription for an initial supply of medications and either dispense the medications directly from the clinician's room or escort the patient to the clinic pharmacy for immediate dispensing, as per the usual practice at each clinic. Patients in the intervention arm will then receive the same token of appreciation (shopping voucher or cash) as was provided earlier to patients in the standard arm.

### Data collection and management
For the primary analysis, the two main sources of study data will be case report forms (CRFs) and patients'

routine medical records. The CRF includes responses to the baseline questionnaire and all four SLATE screens. It also has sections for confirming a participant's randomisation assignment, CD4 count, medication dispensing, next appointment date and/or referral for further care, as needed. It concludes with a follow-up form for recording study outcomes manually from paper files, in case these are not available electronically at every site.

The SLATE CRF will be programmed in REDCap Mobile (https://www.project-redcap.org/software/mobile-app/), housed at Boston University. Research Electronic Data Capture (REDCap) is a secure, web-based application designed to support data capture for research studies.[13] Data capture forms will be built in REDCap's web application and then loaded securely onto tablet computers for direct entry by study staff on site. Using REDCap Mobile, data can be entered off line, then uploaded to the REDCap server as soon as an internet connection is available. The programme will ensure a consistent study identification number for each participant, maintain data security and allow data quality monitoring by the study data management team in near-real time. Tablets will be accessible only to the study staff at the specified site, and data that are uploaded to the server will automatically be erased from the tablets, to reduce any chance of loss of confidentiality.

With the exception of the follow-up form, CRFs will be completed at the end of a patient's enrolment visit. All data for the follow-up period, until all primary and secondary outcomes are reached, will come from routinely collected medical records. All the study sites maintain an electronic medical record (EMR) database that we will access for most follow-up data. In South Africa, sites use Tier.Net, the national HIV monitoring system.[14] In Kenya, sites use either IQCare (https://fgiqcare.codeplex.com/) or the Kenya version of Open MRS (https://wiki.openmrs.org/display/ke/KenyaEMR). Because the electronic records are not always complete or up to date, for example, if a clinic is short-staffed and a data entry backlog develops, the CRF follow-up form will be completed where needed. To fill in missing fields in the EMR databases, we will also search patients' paper files and other clinic data sources (eg, registers) as needed.

To allow linking of data between the CRF and medical records, a separate linking form, also in REDCap, will be completed for each patient immediately after consent. The linking form will contain the patient's name, date of birth, clinic file number(s) and national identification number. It will be stored separately from all other data collection forms and used solely to select the right medical records (electronic and/or paper) for each study participant.

In addition to the two main sources of data described above, data from screening forms, which will be entered into a separate REDCap database, will be used to compare the enrolled sample with all patients screened for the study, to determine if the study sample is representative of the screened population and summarise reasons for screening out of the study. We will also collect aggregate, facility-level information on patient volume, resource utilisation and unit costs to answer our secondary research questions.

CRF data and other information collected in REDCap (screening forms and linking forms) will be uploaded to the REDCap server on a daily basis, using a cellular network uplink from the tablets. Following an initial period of daily quality review, a study manager will monitor data quality and completeness on a weekly basis. Queries about the data will be sent to study coordinators for follow-up and correction by site level staff, as needed. All study investigators (principal and coinvestigators) will have access to the full study data set, which will be centrally managed by the overall study manager, who is also a coinvestigator. The exception to this is the linking form containing individual identifiers, to which only the overall principal investigator, country principal investigators and overall study manager will have access.

Data generated by the study (case report forms) will be made available in deidentified format following closure of the protocol in a publicly available repository, to be identified in papers published from the study. Data obtained from the study sites (routinely generated medical record data) will not be owned by the study and cannot be made publicly available by the authors.

## Data analysis

The analytic approach will be by intention-to-treat: subjects will be analysed according to the intervention they were allocated to receive, whether or not they adhered to the defined intervention. Outcomes for patients randomised to the intervention arm who screen out of immediate ART initiation by the SLATE algorithm will count towards the intervention arm. Analyses will be pooled across countries, but we will also stratify the analysis of primary outcomes by country.

The analysis will begin with a simple comparison of the two treatment groups with respect to baseline predictors of outcomes to detect any imbalances. We will then compare the proportion of patients achieving each dichotomous outcome by group and will calculate crude risk ratios and crude risk difference and their corresponding 95% CIs. We will look for effect modification by important predictors of the outcome. The primary modifier is expected to be country, as standard of care for ART initiation differs between the two study countries. We will also look for differences in effects stratified by age, sex, baseline body mass index, and CD4 count, which are predictors of retention in care, and by any other important demographic, socioeconomic and clinical predictors of outcomes identified, using baseline data including questionnaire responses. Our analysis for effect modification will use a simple stratification of the primary analysis by the potential modifier and reporting crude risk differences and risk ratios and their corresponding 95% CIs.

Analytic methods for secondary outcomes are described in table 2.

Results of the study will be disseminated through papers reporting on primary and secondary outcomes and published in peer-reviewed journals. In addition, shorter presentations of study findings will be prepared for dissemination to the study sites and patients and for use by local policymakers and programme managers. As we expect the results to be of particular relevance to HIV care and treatment guidelines committees both nationally and internationally, efforts will be made to ensure that these bodies have access to the findings.

## Limitations

We anticipate that the SLATE study will have four main limitations. First, the number of study countries will be small, which could reduce generalisability if patient populations differ in characteristics that affect study outcomes. Second, in order to reach the target sample size in the desired timeframe, there will be heterogeneity in the population enrolled, with some patients who have spent years in pre-ART monitoring and others who were diagnosed with HIV on the day of study enrolment. Third, ART-eligible patients who visit the study clinics but are determined by study staff to be too emotionally distraught (eg, by just having learnt of their HIV diagnosis) or physically ill to be asked to participate in a study will be excluded, which may bias the study sample towards patients who are physically or emotionally healthier than the overall population. And fourth, as in most operational research studies, we will have little control over what happens in our standard care (non-intervention) arm. Standard of care continues to evolve rapidly in Africa. Guideline revisions are frequent, and clinicians at the study sites may learn from the intervention and make adjustments to routine care before the study is completed. Any changes that occur will be reported by study staff and taken into account in data analysis. We note that although patients participating the study will receive a payment that will presumably not be offered to patients if the SLATE intervention were adopted into standard care, the payment will be offered to both study arms and should therefore not affect either study outcomes or the generalisability of findings.

## DISCUSSION

Improving the efficiency of HIV service delivery—which encompasses both improving outcomes and reducing costs—is a high priority of national governments and international agencies tasked with implementing and paying for universal access to ART. Patients who have already volunteered for HIV testing but not yet started ART are a promising target for intervention, as they have already identified themselves and, by coming forward for testing, provided a time and place to intervene. Streamlining procedures for HIV initiation thus offers a relatively easy way to increase efficiency. The resources freed by such improvements can, in turn, be used to expand access to and quality of healthcare provision overall.

The SLATE algorithm that we will evaluate in this study has the potential to reduce the time and resources that both providers and patients must invest in ART initiation, while also diminishing the likelihood that patients will get lost from care between diagnosis and treatment initiation. If it can do so without jeopardising outcomes after starting ART—such as retention in care during the first year, as will be monitored by the study—then SLATE will offer national HIV programmes and providers a new and more efficient approach to an important component of HIV care. As other researchers have argued, rigorous implementation studies of feasible and pragmatic approaches to reducing losses to care are essential to achieving global targets.[15] The SLATE study aims to generate evidence for one new such approach.

**Author affiliations**
[1]Department of Global Health, Boston University School of Public Health, Boston, Massachusetts, USA
[2]Health Economics and Epidemiology Research Office, Department of Internal Medicine, School of Clinical Medicine, Faculty of Health Sciences, University of the Witwatersrand, Johannesburg, South Africa
[3]Department of Epidemiology, Boston University School of Public Health, Boston, Massachusetts, USA
[4]Kenya Medical Research Institute/Walter Reed Project HIV Program, Kericho, Kenya
[5]Bill & Melinda Gates Foundation, Seattle, Washington, USA
[6]Wits Reproductive Health and HIV Institute, Department of Internal Medicine, School of Clinical Medicine, Faculty of Health Sciences, University of the Witwatersrand, Johannesburg, South Africa

**Contributors** Conceived of and designed the work: SR, MPF, BAL, PDE and WDFV. Contributed to designing the work: ATB, MM, IT and MB. Wrote the first draft of the manuscript: SR. Reviewed and revised the manuscript: all authors. Approve the manuscript's results and conclusions: all authors. All authors have read, and confirm that they meet, ICMJE criteria for authorship.

**Funding** Funding for the work presented here was provided by the Bill & Melinda Gates Foundation under the terms of OPP1136158 to Boston University. The funders participated in the Technical Consultation reported here and are coauthors of the manuscript. The funders had no separate role, beyond that of other participants and authors, in study design, data collection and analysis, decision to publish or preparation of the manuscript. The views expressed are those of the authors and should not be construed to represent the positions of the US Army or the Department of Defense.

**Competing interests** PDE is an employee of the Bill & Melinda Gates Foundation, which is funding this work. WDFV sits on antiretroviral initiation guideline committees, both local and international. WDFV has accepted speaking honoraria from multiple manufacturers of antiretrovirals and is on several of their advisory boards. The remaining authors declare that they have no competing interests.

**Patient consent** Not applicable.

**Ethics approval** Boston University Institutional Review Board.

**Provenance and peer review** Not commissioned; externally peer reviewed.

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
