## [Reviewer comments · BMJ Open]

ARTICLE DETAILS

TITLE (PROVISIONAL)	Simplified clinical algorithm for identifying patients eligible for immediate initiation of antiretroviral therapy for HIV (SLATE): protocol for a randomized evaluation
AUTHORS	Rosen, Sydney; Fox, Matt; Larson, Bruce; Brennan, Alana; Maskew, Mhairi; Tsikhutsu, Isaac; Bii, Margaret; Ehrenkranz, Peter; Venter, WD Francois

VERSION 1 - REVIEW

REVIEWER	Peter MacPherson University of Liverpool/Liverpool School of Tropical Medicine, United Kingdom
REVIEW RETURNED	28-Feb-2017

GENERAL COMMENTS	Thank you for asking me to review this manuscript. Rosen and colleagues present the protocol for an individually-randomised trial to evaluate the effectiveness of a simplified clinical screening algorithm for improving the timeliness of antiretroviral therapy initiation in HIV-positive adults in South Africa and Kenya. This study is important and leads clearly from the investigators substantial body of influential research in this field. The findings of the trial will be clearly beneficial to clinicians and policymakers who are currently grappling with how best to meet the ambitious WHO “treat all” HIV goals. Before the protocol is published, I have a small number of issues that the investigators should address. 1. It would be worthwhile expanding upon the reason for selecting the primary endpoint of the proportion of participants who initiate ART within 28 days. For a trial that aims to evaluate the effectiveness of an “immediate” ART initiation strategy, this seems a little unambitious. I wonder if there might be a comparable proportion of participants within each group who remain unable to initiate ART after 28 days, diluting estimates of effectiveness. One way to address this might be to provide greater justification for why the primary endpoint is not a comparison between groups of the time to ART treatment initiation (currently a secondary trial outcome).2. Linked to Point 1 above, the authors should provide a clearer definition the timing of intervention delivery, and definitions of participant follow-up time. It is not clear currently in the manuscript whether “immediate ART initiation” refers to initiation on the same day as HIV diagnosis, or on the same day as first ART clinic attendance (there may be important delays between these timepoints if HIV testing is done distal from the ART clinic).3. Further, in the limitations, the authors should add to the discussion to consider whether there is sufficient equipoise in the
--

study design. It seems unlikely that participants in the intervention group would fair worse than those in the control group.

4. Moreover, if the trial showed reduced time to ART initiation in the intervention group, this could be indicative of inappropriately early ART initiation. To counter this, the authors should consider reporting, as secondary outcomes, the proportion of participants in each group who experienced adverse events and serious adverse events, and the timing of such events.

5. Screening and enrolment: It is not clear whether the investigators plan to assess all individuals attending the study clinics during the study period for eligibility, or recruit a fixed or variable number per day if their capacity for recruitment is exceeded. Could they provide a little clarity here, and expand upon how any potential selection bias will be minimised.

6. Could the authors expand upon how they will distinguish between implementation failure and non-effectiveness. As little follow-up of participants is undertaken, it may be that participants allocated to the intervention group do not receive the SLATE screening tool, or are not referred for further assessment in accordance with the protocol. Some description of the planned quality assurance activities around intervention implementation would be helpful.

7. SLATE algorithm (Table 1): clarify that a “positive” TB symptom screen should comprise of the presence of any of these four symptoms.

8. In the Discussion, the authors should consider the possible effects of offering a financial contribution for participation, and whether this may limit generalisability of results (is not standard of care in the region).

9. SLATE algorithm (Table 1): I wonder whether previous adverse reaction to ART would be an additional important screen to include under the “Medical History” component?

10. Page 12, Paragraph 2: Could the authors provide a greater description of how CD4 cell count results will be handled. Will these be provided to participants and clinicians at the study facilities? Whilst I appreciate that CD4 cell count measurement is no longer standard of care, and will not be undertaken in participants allocated to the control group, could the authors comment on whether this intervention may have unintended consequences on the timeliness of ART for intervention group participants (and thus potentially resulting in bias).

11. Statistical analysis: could the authors state whether the trial is powered for the analysis stratified by country? If not, they should consider modifying Lines 32-34 on page 9.

12. For outcome ascertainment, the investigators will rely upon data linkage between CRFs and clinic electronic records. Can they provide some assurance that this linkage is feasible, comprehensive and accurate (perhaps by referencing any previous evaluations of such systems)?

13. Minor suggestion: Page 15, Line 27 – change “supposed to receive” to “allocated to receive”

	14. Minor suggestion: Page 3: Bullet 2 – change “will contribute to achieving global targets...” to “will contribute to the evidence base for achieving global targets...” 15. Minor suggestion: Page 4, Line 56 – change “on the spot” to “on the same-day”.
--	---

REVIEWER	Richard Lessells Clinical Research Fellow Department of Clinical Research London School of Hygiene and Tropical Medicine United Kingdom
REVIEW RETURNED	13-Mar-2017

GENERAL COMMENTS	This is an extremely clear manuscript, which accurately reflects the submitted trial protocol. The research described is certainly important and should be of interest not only to those in the global HIV field but also to those interested more generally in improving the efficiency of health systems. I have just a few minor comments.  1. When describing the standard of care strategy, it would be good to see explicit mention of which cadres of staff currently manage ART in the facilities involved in the trial and how this compares to the intervention arm, and whether there will be any additional training or other support to these staff prior to or during the trial. 2. I couldn't see clearly how the centre effects will be dealt with in the analysis. Presumably it is expected that outcomes may differ by individual site, as well as by country? 3. The description of the sample size calculation is open to a bit of confusion: 'we estimated that 65% of treatment-eligible patients will be initiated on antiretroviral therapy and retained on ART in the standard arm'. Does the 65% just refer to the second primary outcome (initiation within 28d and alive, in care, retained at 8m)? Or does the 65% refer to each of the primary outcomes separately? Ultimately the question is has the study been powered for both primary outcomes? 4. Exclusion for people 'determined by staff to be physically, mentally or emotionally unable to participate'. This is an unusual statement. Is this really talking about people deemed to lack capacity to give informed consent? Otherwise this seems a bit open to the concern that people will be excluded if there is a sense they might be at risk of having poor outcomes. Is it possible to be more explicit with this exclusion criterion? 5. It would be useful for readers interested in this sort of research to see a brief discussion of why individual randomisation was chosen, as opposed to cluster randomisation or other approach. Was this just for statistical efficiency? Might this create logistical and operational challenges at the trial sites?
--

VERSION 1 – AUTHOR RESPONSE

Reviewer: 1

Reviewer Name: Peter MacPherson

Institution and Country: University of Liverpool/Liverpool School of Tropical Medicine, United Kingdom

Competing Interests: None declared

Thank you for asking me to review this manuscript. Rosen and colleagues present the protocol for an individually-randomised trial to evaluate the effectiveness of a simplified clinical screening algorithm for improving the timeliness of antiretroviral therapy initiation in HIV-positive adults in South Africa and Kenya. This study is important and leads clearly from the investigators substantial body of influential research in this field. The findings of the trial will be clearly beneficial to clinicians and policymakers who are currently grappling with how best to meet the ambitious WHO “treat all” HIV goals. Before the protocol is published, I have a small number of issues that the investigators should address.

We thank Dr. MacPherson for his helpful and thought-provoking comments on our manuscript.

1. It would be worthwhile expanding upon the reason for selecting the primary endpoint of the proportion of participants who initiate ART within 28 days. For a trial that aims to evaluate the effectiveness of an “immediate” ART initiation strategy, this seems a little unambitious. I wonder if there might be a comparable proportion of participants within each group who remain unable to initiate ART after 28 days, diluting estimates of effectiveness. One way to address this might be to provide greater justification for why the primary endpoint is not a comparison between groups of the time to ART treatment initiation (currently a secondary trial outcome).

We thank the reviewer for this thoughtful comment. The main reason for choosing 28 days as the primary endpoint is that the most important potential benefit of the intervention is to reduce loss to care (loss to follow up) between testing HIV-positive and starting ART for patients who are not overtly symptomatic (ie probably have relatively high CD4 counts). We are thus focused on improving outcomes for patients who would have dropped out of care entirely (or for a long period) and not initiated ART at all, or not until they had become much sicker. We have not seen any evidence that, for patients who are not yet symptomatic, starting ART within a day or a week is better than starting within 28 days, in terms of patient health outcomes on ART. This is the reason that uptake of ART within 28 days was the consensus “ART initiation” outcome agreed upon at the consultation mentioned in the manuscript (citation 9)—as long as you do start within a short time (proxied by 28 days), whether it is today or next week or the week after doesn’t really matter. So while we would rather have chosen a more challenging threshold (e.g. “initiation within 7 days”) to maximize the apparent effectiveness of the intervention, it seemed more relevant to service delivery, as well as more conservative from a study design perspective, to go with the longer interval.

Because we agree that 28 days is a relatively generous endpoint, we have included two secondary outcomes that provide the data the reviewer is looking for: ART initiation within 14 days (selected to allow comparability with citation 15); and time to initiation, in days. We will report detailed results for the time to initiation outcome, allowing readers to gauge intervention effectiveness at any interval they deem appropriate.

We have added an explanation of our reasoning for the 28-day outcome the description of primary outcomes on page 7.

2. Linked to Point 1 above, the authors should provide a clearer definition the timing of intervention delivery, and definitions of participant follow-up time. It is not clear currently in the manuscript whether “immediate ART initiation” refers to initiation on the same day as HIV diagnosis, or on the same day as first ART clinic attendance (there may be important delays between these timepoints if HIV testing is done distal from the ART clinic).

We apologize for the lack of clarity on this. We originally intended to describe SLATE as “same-day initiation,” in keeping with what is becoming common usage (see, for example, <https://www.surveymonkey.com/r/CNJTB5>, where the WHO defines same day as starting treatment

“on the same day of a diagnosis of HIV or on the same day that a person comes to a clinic looking for treatment for HIV”). We did not want to imply, however, that a patient who “screens out” on SLATE cannot receive ART on the same day. A patient who just needs a little additional counseling about adherence before starting ART, for example, would screen out under SLATE but could still easily start on the same day if the clinic had a counselor available to provide that service. Some clinicians we have consulted believe that the criteria to “screen in” under SLATE are very conservative and are concerned that we should not imply that patients who screen out must have ART initiation delayed by a substantial period of time.

In view of this, we have tried to clarify in the manuscript two points. First, we emphasized that for the SLATE study, patients can enrol at any time before starting ART, whether it is a few minutes after the patient’s first positive HIV test or after multiple pre-ART care visits (page 11). The reviewer is certainly correct that a patient who had an HIV test at a distant testing site and then presented for treatment at a clinic six months later is very different from a patient who had a test at a clinic and was offered treatment that same day, but both groups are included in the population of patients seeking treatment, and SLATE is relevant to both.

Second, we clarified that in the context of this manuscript, “immediate initiation” refers to being dispensed ARVs right after completing (and screening in on) the four SLATE screens (pp 5-6). We hope that these changes have addressed the reviewer’s concerns.

3. Further, in the limitations, the authors should add to the discussion to consider whether there is sufficient equipoise in the study design. It seems unlikely that participants in the intervention group would fare worse than those in the control group.

We appreciate the reviewer’s concern but strongly believe that there is sufficient equipoise to justify this trial. The SLATE algorithm has never been tried before and, while we certainly hope that subjects in the intervention group will not fare worse than those in the control group, it is possible that the algorithm will perform worse than standard are in identifying patients who require additional services prior to starting ART. As noted in the manuscript, it is also possible that patients who are put onto treatment immediately, under the SLATE algorithm, will be more likely to drop out of care after initiation because the intervention starts them too quickly, without adequate preparation. This latter concern has been raised on multiple occasions and is the main justification for primary outcome 2, which assesses patient outcomes after treatment initiation. While we hypothesize that patients offered the SLATE intervention will do better, we do believe that there is a possibility of them doing worse. Thus we consider there to be sufficient balance to support the need for a trial.

We did not comment on this in the manuscript as it has not been raised before by any of the scientific or ethical reviewers of the protocol and we are aware that the manuscript is already pushing against length limits due to the other additions made in response to reviews. If the editors consider it important, however, we would be glad to incorporate the response above into the paper.

4. Moreover, if the trial showed reduced time to ART initiation in the intervention group, this could be indicative of inappropriately early ART initiation. To counter this, the authors should consider reporting, as secondary outcomes, the proportion of participants in each group who experienced adverse events and serious adverse events, and the timing of such events.

We agree that inappropriate early ART initiation is a possible consequence of the study intervention. For this reason, both our primary and secondary outcomes include comparison of patient status (retention in care and viral suppression) at various intervals after starting ART, and we anticipate describing reasons for not achieving these outcomes (e.g. loss to follow up due to an adverse event) in reporting on these outcomes. Adverse events will of course be reported to the ethics boards and

described post-hoc as part of the overall reporting of study results. Although we are not in a position to add additional outcomes to the protocol at this time, as study recruitment has begun, we will consider this issue, as suggested by the reviewer. We have added a note about this on page 8 and in Table 2.

5. Screening and enrolment: It is not clear whether the investigators plan to assess all individuals attending the study clinics during the study period for eligibility, or recruit a fixed or variable number per day if their capacity for recruitment is exceeded. Could they provide a little clarity here, and expand upon how any potential selection bias will be minimised.

We will screen for study eligibility a variable number of potential subjects per day at each site, based on study staff capacity. Patients who are referred by clinic staff to the study assistant(s) on site will be screened on a first-come, first-served basis. We expect to lose some potentially eligible patients because the study assistants are not available when the patients are, but we do not anticipate that this process will create a selection bias, as the arrival of patients at the study clinics and referral to the study assistant is effectively random. We have added further explanation of this process to the manuscript on page 11.

6. Could the authors expand upon how they will distinguish between implementation failure and non-effectiveness. As little follow-up of participants is undertaken, it may be that participants allocated to the intervention group do not receive the SLATE screening tool, or are not referred for further assessment in accordance with the protocol. Some description of the planned quality assurance activities around intervention implementation would be helpful.

Because this is a randomized controlled trial, we do not anticipate the kinds of problems with implementation failure that are often experienced in large program evaluations. Study staff have been carefully trained on the study protocol and procedures and been observed in conducting these procedures. In addition, the study coordinator makes weekly visits to each study site to review and observe procedures.

Beyond ensuring well-trained and protocol-compliant staff, the REDCap data platform being used to capture data allows daily monitoring of all patient data, including the times at which each step was completed and bar codes to confirm laboratory test samples and results. The tablet-based data entry forms used by the study staff are programmed to prevent inconsistent or incorrect results to be entered, and require that all fields be completed sequentially. Any errors or illogical or suspect entries can be identified in real time by the investigators and addressed.

As is explained in the protocol, we will not interact with study subjects following the enrollment visit and do not have any way to monitor the quality of the routinely collected medical record data we will use for follow up. We will thoroughly collect follow up data from disparate sources on site (e.g. patient files, registers, laboratory results) to create as high quality a follow up data set as possible, but the variable completeness of routinely collected medical record data cannot be avoided in a study like this one.

As the manuscript already comments on data quality monitoring on page 15, and we do not expect the type of implementation failure mentioned by the reviewer due to the study design, we have not added further text to the manuscript in response to this comment. We are happy to do so if the editors advise, however.

7. SLATE algorithm (Table 1): clarify that a “positive” TB symptom screen should comprise of the presence of any of these four symptoms.

Table 1 in the manuscript indicates that “Current cough, fever, night sweats, or recent weight loss” is a reason for screening out under SLATE. We do not understand the reviewer’s request for clarification but would be glad to make a revision if needed. We have adjusted the heading of the last column in Table 1 in the hopes of making it easier to understand.

8. In the Discussion, the authors should consider the possible effects of offering a financial contribution for participation, and whether this may limit generalisability of results (is not standard of care in the region).

The payment for participation is required by local ethics boards for clinical trials and is intended to acknowledge patients’ willingness to take the time to participate. As it is offered to both study arms, we do not expect it to affect trial results. We agree that it may increase willingness to participate in the study itself, but we cannot come up with any reason that it will affect patients’ behavior or outcomes with regard to the SLATE algorithm or ART initiation. We therefore are not concerned that it will limit generalizability of results. We have added a comment about this on pp 17-18.

9. SLATE algorithm (Table 1): I wonder whether previous adverse reaction to ART would be an additional important screen to include under the “Medical History” component?

This would indeed be interesting to know, but because patients who have previously been on ART will screen out of the algorithm in the Medical History screen anyway, there is no need to ask this further question. Our goal in designing the algorithm was to keep it as brief and simple as possible, and we refrained from including questions that are not essential to make the “screen in/screen out” decision.

10. Page 12, Paragraph 2: Could the authors provide a greater description of how CD4 cell count results will be handled. Will these be provided to participants and clinicians at the study facilities? Whilst I appreciate that CD4 cell count measurement is no longer standard of care, and will not be undertaken in participants allocated to the control group, could the authors comment on whether this intervention may have unintended consequences on the timeliness of ART for intervention group participants (and thus potentially resulting in bias).

In both study countries, it remains standard of care to perform a baseline CD4 count on all patients initiating ART. We thus expect to have CD4 count results for patients in both arms of the study. From the study’s perspective, this will enrich the analysis by allowing us to look for associations between CD4 count and study outcomes, as noted in the data analysis section.

In the intervention arm, patients who are not presenting with a prior, recent CD4 test result will not know their own CD4 counts at the time they are offered immediate ART initiation under the SLATE algorithm. We assume they will be told these results at their next routine clinic visit, when the report has come back from the lab. We do not know exactly when patients in the standard of care arm will receive their CD4 count results, but since a CD4 count is no longer required for ART eligibility, we assume that some will learn their CD4 counts before starting ART and some after.

It is possible that patients in the standard arm who learn their CD4 counts before initiating ART will incorporate that information into their decisions about whether to start, and this could affect uptake in that arm. This would not be a study bias, however, but rather one of the differences between the arms that explains the study outcomes. It is an interesting possibility, but it is unlikely that the routinely collected follow-up data will allow us to determine exactly when (before or after starting ART) standard arm patients were given their CD4 count results. We have clarified that baseline CD4 counts will continue to be performed under standard care on page 14.

11. Statistical analysis: could the authors state whether the trial is powered for the analysis stratified

by country? If not, they should consider modifying Lines 32-34 on page 9.

The trial is powered for the analysis stratified by country. We have reiterated this on page 9 in the sample size section, as it is already noted at the end of the first paragraph of the data analysis section.

12. For outcome ascertainment, the investigators will rely upon data linkage between CRFs and clinic electronic records. Can they provide some assurance that this linkage is feasible, comprehensive and accurate (perhaps by referencing any previous evaluations of such systems)?

We will collect individual medical record numbers used by the study clinics as part of the CRF to ensure that we can make this link. As doing the linking completely and accurately is essential to the success of the study, recording all available linking information is a high priority for the study teams on site. We have done this successfully on multiple occasions, most recently for the RapIT trial (citation 14).

13. Minor suggestion: Page 15, Line 27 – change “supposed to receive” to “allocated to receive”

Good suggestion, we have made this change.

14. Minor suggestion: Page 3: Bullet 2 – change “will contribute to achieving global targets...” to “will contribute to the evidence base for achieving global targets...”

We agree and have made this change.

15. Minor suggestion: Page 4, Line 56 – change “on the spot” to “on the same-day”.

For the reason explained above, we do not want to confound “immediate” and “same day” treatment. We have therefore kept “on the spot” to emphasize that under SLATE, there are no more steps to be completed before ARVs are dispensed.

Reviewer: 2

Reviewer Name: Richard Lessells

Institution and Country: Clinical Research Fellow, Department of Clinical Research, London School of Hygiene and Tropical Medicine, United Kingdom

Competing Interests: None declared

This is an extremely clear manuscript, which accurately reflects the submitted trial protocol. The research described is certainly important and should be of interest not only to those in the global HIV field but also to those interested more generally in improving the efficiency of health systems. I have just a few minor comments.

We appreciate that Dr. Lessells believes that the study will be important and thank him for these comments.

1. When describing the standard of care strategy, it would be good to see explicit mention of which cadres of staff currently manage ART in the facilities involved in the trial and how this compares to the intervention arm, and whether there will be any additional training or other support to these staff prior to or during the trial.

In Kenya, clinical officers initiate ART. In South Africa, ART is usually initiated by public health nurses. These same cadres of clinicians will initiate ART under SLATE in the intervention arm. They will be

trained in study procedures but not receive any additional clinical or other training that would differentiate them from typical public sector providers. We have added a more detailed description of staff cadres and training on page 10.

2. I couldn't see clearly how the centre effects will be dealt with in the analysis. Presumably it is expected that outcomes may differ by individual site, as well as by country?

While the trial is being conducted at multiple sites, it is randomized at the level of the individual patient, not cluster-randomized. There is thus no need to adjust for clustering by site. The study is only powered for analysis by country, and we will look for both site- and country-specific differences. We will also investigate differences by basic demographic characteristics (e.g. sex), though the study is not specifically powered to detect all these differences. Where multivariate analysis is used, we will incorporate site fixed effects as appropriate.

3. The description of the sample size calculation is open to a bit of confusion: 'we estimated that 65% of treatment-eligible patients will be initiated on antiretroviral therapy and retained on ART in the standard arm'. Does the 65% just refer to the second primary outcome (initiation within 28d and alive, in care, retained at 8m)? Or does the 65% refer to each of the primary outcomes separately? Ultimately the question is has the study been powered for both primary outcomes?

The study was powered for the second primary outcome. We have clarified this in the sample size description on page 9.

4. Exclusion for people 'determined by staff to be physically, mentally or emotionally unable to participate'. This is an unusual statement. Is this really talking about people deemed to lack capacity to give informed consent? Otherwise this seems a bit open to the concern that people will be excluded if there is a sense they might be at risk of having poor outcomes. Is it possible to be more explicit with this exclusion criterion?

This exclusion is intended to account for the fact that many study-eligible patients will either have just received the news that they are HIV-positive and be very upset ("emotionally unable to participate") or have come to the clinic because they are very sick and do not feel well enough to sit through the consent process, questionnaire, and other procedures ("physically unable to participate"). There may also be patients who lack the intellectual capacity to give informed consent ("mentally unable to participate"). Allowing exclusion for all these reasons is ethically necessary if one wishes to recruit in an HIV clinic, and we have no choice but to rely on the judgment of our site-level study staff to determine if a patient should be excluded for one of these reasons.

We agree with the reviewer that this exclusion criterion has the potential to bias the sample, not because patients will deliberately be excluded because of the risk of poor outcomes, but because the sample enrolled will not be representative of all ART-eligible patients presenting at the clinic. It will be biased in favor of patients who are able to participate. This should certainly be noted in the manuscript, and we thank the reviewer for bringing it up. We have added it to the limitations section on page 17.

5. It would be useful for readers interested in this sort of research to see a brief discussion of why individual randomisation was chosen, as opposed to cluster randomisation or other approach. Was this just for statistical efficiency? Might this create logistical and operational challenges at the trial sites?

This is an interesting question and we appreciate the reviewer mentioning it. We did consider cluster randomization, but the intervention is well-suited to individual randomization, as it does not require

facility-level health systems changes. Since the intervention simply alters the procedures followed by individual patients and clinicians, randomization at the individual level does not create any particular logistical or operational challenges. Contamination between arms seems unlikely based on the structure of the study and the logistics of patient flow and staff activities at the sites. Since we did not face problems with either implementation or contamination, individual randomization was considered to be a stronger design and, as suggested by the reviewer, more efficient in terms of enrollment numbers and resource utilization.

We have not added this to the manuscript because we are aware that the manuscript is already pushing against length limits due to the other additions made in response to reviews, and we have tried to avoid topics that are interesting in general but not high priority in understanding this protocol. If the editors consider it important, however, we would be glad to incorporate the response above into the paper.

VERSION 2 – REVIEW

REVIEWER	Peter MacPherson Liverpool School of Tropical Medicine, UK
REVIEW RETURNED	27-Mar-2017

GENERAL COMMENTS	Thank you. The authors have fully addressed all the questions raised. Good luck with an important study!
--

REVIEWER	Richard Lessells London School of Hygiene and Tropical Medicine United Kingdom
REVIEW RETURNED	23-Mar-2017

GENERAL COMMENTS	The authors have comprehensively addressed the comments from me and from the other reviewer. The changes to the manuscript have been positive. I am comfortable that there are a couple of issues that were acknowledged but did not result in manuscript changes.
--